# *Enterobacter cloacae* infection characteristics and outcomes in battlefield trauma patients

**William Bennett**[1,2]*, **Katrin Mende**[1,3,4], **Wesley R. Campbell**[5], **Miriam Beckius**[1], **Laveta Stewart**[3,4], **Faraz Shaikh**[3,4], **Azizur Rahman**[3,4], **David R. Tribble**[3], **Joseph M. Yabes**[1,2]

1 Brooke Army Medical Center, JBSA Fort Sam Houston, Texas, United States of America, 2 Uniformed Services University of the Health Sciences, Bethesda, MD, United States of America, 3 Infectious Disease Clinical Research Program, Department of Preventive Medicine and Biostatistics, Uniformed Services University of the Health Sciences, Bethesda, Maryland, United States of America, 4 Henry M. Jackson Foundation for the Advancement of Military Medicine, Inc., Bethesda, Maryland, United States of America, 5 Walter Reed National Military Medical Center, Bethesda, Maryland, United States of America

* william.n.bennett.mil@health.mil

## Abstract

*Enterobacter cloacae* is a Gram-negative rod with multidrug-resistant potential due to chromosomally-induced AmpC β-lactamase. We evaluated characteristics, antibiotic utilization, and outcomes associated with battlefield-related *E. cloacae* infections (2009–2014). Single initial and serial *E. cloacae* isolates ($\geq$24 hours from initial isolate from any site) associated with a clinical infection were examined. Susceptibility profiles of initial isolates in the serial isolation group were contrasted against last isolate recovered. Characteristics of 112 patients with *E. cloacae* infections (63 [56%] with single initial isolation; 49 [44%] with serial isolation) were compared to 509 patients with bacterial infections not attributed to *E. cloacae*. *E. cloacae* patients sustained more blast trauma (78%) compared to non-*E. cloacae* infections patients (75%; p<0.001); however, injury severity scores were comparable (median of 34.5 and 33, respectively; p = 0.334). Patients with *E. cloacae* infections had greater shock indices (median 1.07 vs 0.92; p = 0.005) and required more initial blood products (15 vs. 14 units; p = 0.032) compared to patients with non-*E. cloacae* infections. Although *E. cloacae* patients had less intensive care unit admissions (80% vs. 90% with non-*E. cloacae* infection patients; p = 0.007), they did have more operating room visits (5 vs. 4; p = 0.001), longer duration of antibiotic therapy (43.5 vs. 34 days; p<0.001), and lengthier hospitalizations (57 vs. 44 days; p<0.001). Patients with serial *E. cloacae* had isolation of infecting isolates sooner than patients with single initial *E. cloacae* (median of 5 vs. 8 days post-injury; p = 0.046); however, outcomes were not significantly different between the groups. Statistically significant resistance to individual antibiotics did not develop between initial and last isolates in the serial isolation group. Despite current combat care and surgical prophylaxis guidelines recommending upfront provision of AmpC-inducing antibiotics, clinical outcomes did not differ nor did significant antibiotic resistance develop in patients who experienced serial isolation of *E. cloacae* versus single initial isolation.

**Data Availability Statement:** All relevant data are contained in the paper. Data for this study are available from the Infectious Disease Clinical Research Program (IDCRP), headquartered at the

USU, Department of Preventive Medicine and Biostatistics. Review by the USU Institutional Review Board and approval of data sharing agreements are required for use of the data collected under this protocol. Data requests may be sent to: Address: 6270A Rockledge Drive, Suite 250, Bethesda, MD 20817. Email: contactus@idcrp.org.

**Funding:** Support for this work (IDCRP-024) was provided by the Infectious Disease Clinical Research Program (IDCRP), a Department of Defense program executed through the Uniformed Services University of the Health Sciences, Department of Preventive Medicine and Biostatistics through a cooperative agreement with The Henry M. Jackson Foundation for the Advancement of Military Medicine, Inc. (HJF). This project has been funded by the National Institute of Allergy and Infectious Diseases, National Institutes of Health, https://www.niaid.nih.gov/, under Inter-Agency Agreement Y1-AI-5072 to DRT, the Defense Health Program, U.S. DoD, under award HU0001190002 to DRT, the Department of the Navy under the Wounded, Ill, and Injured Program (HU0001-10-1-0014) to DRT, and the Military Infectious Diseases Research Program, https://midrp.amedd.army.mil/ (HU0001-15-2-0045) to KM. The funders had no role in study design, data collection and analysis, decision to publish, or preparation of the manuscript. Support in the form of salaries was provided by HJF for authors KM, LS, FS, AR; HJF did not have any additional role in the study design, data collection and analysis, decision to publish, or preparation of the manuscript. The specific roles of these authors are articulated in the 'author contributions' section.

**Competing interests:** I have read the journal's policy and the authors of this manuscript have the following competing interests: KM, LS, FS, and AR are/were employees of the Henry M. Jackson Foundation for the Advancement of Military Medicine, Inc. (HJF), a not-for-profit Foundation authorized by Congress to support research at the Uniformed Services University of the Health Sciences (USU) and throughout military medicine. This does not alter our adherence to PLOS ONE policies on sharing data and materials. Please see Data Availability Statement.

## Introduction

The microbiology of infections associated with combat-related injuries have transitioned to a predominance of Gram-negative bacilli since the Vietnam War, as early wound debridement and anatomic site-directed empiric antibiotics have reduced the number of infections secondary to Gram-positive organisms [1]. In the U.S. Naval Hospital in DaNang, Vietnam, 52% of severe extremity wound cultures prior to debridement grew Gram-negative organisms and 20% of the entire wounded cohort developed *Enterobacter* spp. bacteremia by the fifth day of hospitalization [2]. During military operations in Afghanistan, there was a similar prevalence (56%) of Gram-negative organisms from mangled lower extremities on pre-operative wound cultures [3]. Both theaters of war saw wound cultures trend toward a Gram-negative predominance as hospitalization progressed [4]. As modern combat-related injuries have led to a surge in Gram-negative hospital-related infections, multidrug-resistant Gram-negative (MDRGN) infections have become an increasing threat [5].

Microbiology of wounds and wound infections among blast casualties injured in Iraq and Afghanistan has been described, and a higher prevalence of MDR organisms was found in polymicrobial infections compared to monomicrobial infections [6]. In that analysis, *Enterobacter* spp. was the second most commonly isolated Gram-negative organism from polymicrobial cultures and was noted to have a shorter time from injury to first infection. As polymicrobial infections among blast wound infections were more likely to produce MDR organisms and *Enterobacter* spp. infections have a potentially quicker onset, earlier identification and treatment of *Enterobacter* spp. infections may improve patient outcomes.

As MDRGN infections were being recognized as a worsening threat to patients in military hospitals, U.S. civilian hospitals saw a similar rise in these challenging infections. In the early 2000s, an epidemic of carbapenem-resistant *Enterobacterales* spread across the northeastern United States [7]. A subsequent analysis by the Veterans Health Administration from 2006–2015 indicated that *Enterobacter cloacae* was one of the emerging pathogens with dramatically increasing resistance rates largely secondary to AmpC β-lactamase induction [8]. *E. cloacae* is initially phenotypically susceptible to 3rd generation cephalosporins (e.g., cefotaxime, ceftriaxone, and ceftazidime) *in vitro*; however, as high as 19% of isolates developed β-lactam resistance during treatment [9–12]. Making matters more difficult, clinical laboratories do not typically test for AmpC production, and molecular testing is required to differentiate between chromosomally-induced or constitutively-expressed plasmid AmpC [9]. Third-generation cephalosporins are also not the only AmpC inducers, as amoxicillin / clavulanic acid, cefoxitin, 1st generation cephalosporins, and carbapenems are similarly potent inducers [13, 14].

Guidance surrounding the treatment of *E. cloacae* focuses primarily on bacteremia secondary to respiratory, urinary, intravascular, or intra-abdominal sources, but largely neglects wound infections [15, 16]. Due to concerns regarding potential delays in transport/medevac of combat casualties, the Department of Defense (DoD) Tactical Combat Casualty Care (TCCC) guidelines recommend use of ertapenem for care of open traumatic wounds in the field [17]. In addition, the DoD Joint Trauma System (JTS) clinical practice guidelines, as well as non-DoD guidelines, recommend cefazolin for post-injury prophylaxis [18, 19]. As both ertapenem and cefazolin may induce AmpC production, there is concern of increasing β-lactam resistance negatively affecting patient outcomes in infections where *E. cloacae* is the primary pathogen. Herein, we assessed the epidemiological characteristics of patients with *E. cloacae* infections, characterized the antibiotic prescribing patterns used in the treatment of these infections and their effects on developing resistance, and examined clinical outcomes compared to battlefield trauma patients who developed bacterial infections associated with organisms other than *E. cloacae*. We described clinical characteristics and outcome differences

between patients with single initial isolation of *E. cloacae* and patients with serial isolation to assess for antimicrobial resistance development.

## Materials and methods

### Study population and definitions

Data and specimens were collected through the DoD–Veterans Affairs Trauma Infectious Disease Outcomes Study (TIDOS), which is an observational, longitudinal study of infectious outcomes among military personnel who were wounded in Iraq or Afghanistan (2009–2014) [20, 21]. Criteria for inclusion in TIDOS were being ≥18 years of age, active-duty personnel or DoD beneficiaries injured during deployment, and medically evacuated to Landstuhl Regional Medical Center in Germany with subsequent transfer to a participating military hospital in the United States. The participating U.S. military hospitals were Brooke Army Medical Center (BAMC) in San Antonio, TX, and Walter Reed National Military Medical Center in the National Capital Region (NCR) (prior to September 2011 it was National Naval Medical Center and Walter Reed Army Medical Center). Patients were included if they had *E. cloacae* isolation associated with a clinical infection diagnosis. Patients with a clinical diagnosis of a bacterial infection attributed to an organism(s) other than *E. cloacae* comprised the comparator population for the analysis. The Institutional Review Board (IRB) of the Uniformed Services University of the Health Sciences (USU, Bethesda, MD) approved this study. Data and specimens were collected from individuals who provided authorization through informed consent and HIPAA authorization processes, or through an IRB-approved waiver of consent for use of de-identified data not obtained through interaction or intervention with human subjects.

Demographics, injury characteristics, and early casualty care data were obtained from the DoD Trauma Registry (DoDTR). Infection-related data (e.g., infection syndromes, microbiology, and antibiotic management) were collected from the TIDOS Infectious Disease module of the DoDTR [22]. Data on use of ertapenem in the prehospital setting were not collected by the DoDTR. Tetracycline use was excluded from the analysis as doxycycline was prescribed to military personnel deployed to Afghanistan for antimalarial prophylaxis and continued for 28 days following departure from the country, per DoD guidelines.

Infections were identified using a combination of clinical (e.g., signs and symptoms from direct observations) and laboratory (e.g., microbiology) findings, and classified based on National Healthcare Safety Network definitions, as previously described [20, 23]. Isolates recovered during workups for clinical infection were classified as infecting. Inclusion criteria for the single initial *E. cloacae* isolate group required patients to have an infecting *E. cloacae* isolate collected from the initial culture (may be either monomicrobial or polymicrobial) with no isolation from subsequent cultures. For the serial *E. cloacae* isolate group, all patients with multiple non-colonizing *E. cloacae* isolates cultured at least one day apart were included, as prior studies have determined that *E. cloacae* may develop resistance to β-lactams after one day of therapy [10]. If multiple isolates were collected on the first day of *E. cloacae* isolation, the isolates collected from sterile body sites (more likely to be a true infection) or more proximal wound sites (wounds less likely to undergo early amputation) were given preference. In addition, all isolates must have been stored in the TIDOS specimen repository.

### Laboratory analysis

Identification and susceptibility testing of the *E. cloacae* isolates were performed using the BD Phoenix Automated Microbiology System (NMIC/ID-308 and NMIC-311 panels, BD Diagnostics, Sparks, MD). Antimicrobial susceptibility testing results were interpreted in

accordance with the Clinical Laboratory Standards Institute (CLSI M100 30[th] edition) break-points to construct an antibiogram [24]. Initial and last isolates cultured from patients in the serial isolate group were compared to assess the changing resistance patterns and we regarded any isolates with intermediate susceptibility as resistant. As molecular assays to assess for AmpC were not routinely conducted at the military hospitals during the study period, data on AmpC induction were not available.

## Statistical analysis

Patients with *E. cloacae* infections were analyzed against the comparator population of patients with non-*E. cloacae* infections. Characteristics of *E. cloacae* patients with single initial or serial isolates and characteristics were compared. Categorical variables were assessed using $X^2$ and Fisher's Exact Tests, where appropriate. Continuous variables were analyzed using Mann-Whitney U. Statistical analysis was performed using IBM SPSS Statistics 22 (Version 22 IBM, NY, 2013.). A *p* value of <0.05 was considered statistically significant.

# Results

## Population characteristics

A total of 112 patients with infections due to *E. cloacae* and 509 patients with non-*E. cloacae* infections met inclusion criteria for the analysis. The majority of patients were young males with median age of 24 years (interquartile range [IQR] 21–28) in the U.S. Army who suffered blast injuries from improvised explosive devices while on foot patrol in Afghanistan (Table 1). Patients diagnosed with *E. cloacae* infections sustained more blast injuries (89% vs 75%; p = 0.001) and burns (16% vs. 9%; p = 0.027), had higher first documented shock indices (1.07 vs 0.92; p = 0.005), received more blood products within the first 24 hours of injury (15 vs. 14 units; p = 0.032), and required a greater number of visits to the operating room (5 vs. 4; p<0.001; Table 1).

Ninety percent of the 621 patients in the population sustained extremity injuries with the patients with *E. cloacae* infections having a greater proportion compared to the non-*E. cloacae* infected patients (98% vs. 88%; p = 0.016; Table 1). Despite the higher initial shock indices and blood product requirements, there was no significant difference in the injury severity scores between *E. cloacae* and non-*E. cloacae* infected patients (35 vs. 33; p = 0.33). A higher proportion of patients admitted to BAMC developed a *E. cloacae* infection (32% vs 19% among patients with non-*E. cloacae* infections; p = 0.004). Although fewer patients with *E. cloacae* infections required mechanical ventilation (64% vs. 73%; p = 0.03) or intensive care unit (ICU) admission (80% vs. 90%; p = 0.007), they did have longer hospitalizations (57 vs. 44 days; p<0.001). Patients with *E. cloacae* infections received significantly more total days of antibiotic therapy (43.5 vs. 34 days p <0.001). These patients also were treated significantly more days with carbapenems, 1[st] generation cephalosporins, fluoroquinolones, and vancomycin (Table 2). There was no significant mortality difference between the groups and 97% of the total population survived (Table 1).

*E. cloacae* isolates were linked to 49 patients with serially infecting cultures and 63 patients with single initial infecting cultures. The patients with single initial and serial *E. cloacae* isolation were of similar median age (23 and 24 years, respectively), with the majority sustaining blast injuries (85.7% and 93.9%, respectively) resulting in a minority of burn wounds (15.9% and 16.3%, respectively) and a similar proportion of ICU admissions (81% and 79.6%, respectively; Table 3). There was a higher proportion of polymicrobial infections among the patients who had serial isolates compared to single initial isolates (86% and 67%, respectively; p = 0.021). Single initial vs serial isolation was not associated with a difference in number of

**Table 1. Characteristics of patients with and without *Enterobacter cloacae* infection.**

| Characteristic, No. (%) | Patients with *E. cloacae* infection (N = 112) | Patients with non-*E. cloacae* infection (N = 509)[a] | All Patients (N = 621) | p-value |
|---|---|---|---|---|
| Age at injury, median (IQR) | 23 (21–27) | 24 (22–29) | 24 (21–28) | 0.065 |
| Male | 111 (99.1) | 502 (98.6) | 613 (98.7) | 1.000 |
| *Branch of Service* | | | | 0.449 |
| Air Force and Navy | 5 (4.5) | 36 (7.1) | 41 (6.6) | |
| Army | 68 (60.7) | 298 (58.5) | 366 (58.9) | |
| Marine | 39 (34.8) | 168 (33.0) | 207 (33.3) | |
| Other | 0 (0) | 7 (1.4) | 7 (1.1) | |
| *Combat Theater* | | | | 0.069 |
| Afghanistan | 108 (96.4) | 460 (90.4) | 568 (91.5) | |
| Iraq | 4 (3.6) | 31 (6.1) | 35 (5.6) | |
| Non-theater | 0 (0) | 18 (3.5) | 18 (2.9) | |
| Combat Injury | 108 (96.4) | 476 (93.5) | 584 (94.0) | 0.239 |
| *Mechanism of Injury* | | | | **0.001** |
| Blast | 100 (89.3) | 383 (75.2) | 483 (77.8) | |
| Non-blast | 12 (10.7) | 126 (24.8) | 138 (22.2) | |
| *Blast Type* | | | | 0.410 |
| IED | 93 (83.0) | 346 (68.0) | 439 (70.7) | |
| Non-IED | 7 (6.2) | 37 (7.2) | 44 (7.1) | |
| Injured on foot patrol | 76 (67.9) | 286 (56.2) | 362 (58.3) | 0.213 |
| Burn | 18 (16.1) | 46 (9.0) | 64 (10.3) | **0.027** |
| 1st documented shock index, median (IQR) | 1.07 (0.74–1.49) | 0.92 (0.70–1.22) | 0.93 (0.70–1.27) | **0.005** |
| 1st 24 hour blood transfusion, median units (IQR) | 15 (8–31) | 14 (6–24) | 14 (6–25) | **0.032** |
| *Body Region of Injury* | | | | **0.016** |
| Lower extremity | 21 (18.7) | 89 (17.5) | 110 (17.7) | |
| Upper extremity | 7 (6.2) | 26 (5.1) | 33 (5.3) | |
| Both lower and upper extremity | 82 (73.2) | 334 (65.6) | 416 (67.0) | |
| Non extremity | 2 (1.8) | 60 (11.8) | 62 (10.0) | |
| Injury severity score, median (IQR) | 34.5 (24–45) | 33 (24–43) | 33 (24–43) | 0.334 |
| U.S. military hospital | | | | **0.004** |
| BAMC | 36 (32.1) | 94 (18.5) | 130 (20.9) | |
| NCR | 73 (65.2) | 389 (76.4) | 462 (74.4) | |
| Both BAMC and NCR | 3 (2.7) | 26 (5.1) | 29 (4.7) | |
| *Mechanical ventilation* | | | | **0.030** |
| LRMC only | 16 (14.3) | 126 (24.7) | 142 (22.9) | |
| LRMC & U.S. hospital ≤1 week | 55 (49.1) | 242 (47.5) | 297 (47.8) | |
| LRMC & U.S. hospital ≥2 weeks | 3 (2.7) | 4 (0.8) | 7 (1.1) | |
| None | 38 (33.9) | 137 (26.9) | 175 (28.2) | |
| ICU admission | 90 (80.4) | 456 (89.6) | 546 (87.9) | **0.007** |
| Number of operating room visits, median (IQR) | 5 (4–6) | 4 (3–6) | 5 (3–6) | **<0.001** |
| Hospitalization, median days (IQR) | 57 (40.5–84.5) | 44 (30–62) | 45 (33–66) | **<0.001** |
| Death | 5 (4.5) | 12 (2.4) | 17 (2.7) | 0.216 |

BAMC–Brooke Army Medical Center; ICU–intensive care unit; IED–improvised explosive device; IQR–interquartile range; LRMC–Landstuhl Regional Medical Center; NCR–National Capital Region

[a] Predominant non-*E. cloacae* infections include coagulase-negative staphylococci (13%), *Pseudomonas aeruginosa* (12%), *Escherichia coli* (10.5%), *Acinetobacter calcoaceticus-baumannii* complex (8%), and *Enterococcus faecium* (8%).

**Table 2. Total duration of antibiotic use among patients with and without *E. cloacae* infections[a].**

| Antimicrobials | Duration of Antibiotic Use, median days (IQR) | | | |
|---|---|---|---|---|
| | Patients with *E. cloacae* infection (N = 112) | Patients with non-*E. cloacae* infection (N = 509) | All Patients (N = 621) | p-value |
| Aminoglycoside | 1 (0–7) | 1 (0–4) | 1 (0–4) | 0.256 |
| Carbapenem | 12.5 (4–21.5) | 9 (1–18) | 9 (2–18) | **0.003** |
| Cephalosporin- 1st generation | 8.5 (4–15) | 7 (3–12) | 7 (4–13) | **0.018** |
| Fluoroquinolone | 9 (1.5–15.5) | 5 (0–11) | 5 (1–13) | **0.003** |
| Vancomycin | 10 (2–27) | 0 (0–0) | 0 (0–0) | **<0.001** |
| Total antibiotic duration[b] | 43.5 (32.5–71.0) | 34 (24–50) | 36 (26–52) | **<0.001** |

IQR–interquartile range

[a] Antibiotics that were used for a median of zero days in both groups are not shown and include aminopenicillin, anti-pseudomonal penicillin, 2nd generation cephalosporin, 3rd generation cephalosporin, 4th generation cephalosporin, clindamycin, linezolid, macrolide, monobactam, penicillin, penicillinase-resistant penicillin, polymyxin, trimethoprim-sulfamethoxazole, and topical antibiotic therapy.

[b] Total antibiotic duration was calculated as the total number of days at least one antibiotic was administered. Any days on which no antibiotics were administered are not counted in this measure.

operating room visits (median of 5 for both groups), length of hospitalization (median of 57 days for both groups), or death (5% and 4%, respectively; Table 3). Although there was not a significant difference in the duration of antibiotic therapy (median of 39 and 47 days for single initial and serial isolation respectively), there was a trend toward greater 1st generation cephalosporin utilization in patients who experienced serial isolation of *E. cloacae* (median of 11 vs. 7 days with single initial isolation; p = 0.052; Table 4); however, this does not control for duration of hospitalization and number of visits to the operating room, which would drive use of 1st generation cephalosporins in these trauma patients.

Among the 84 patients with polymicrobial infections, *Pseudomonas aeruginosa* was the most frequently isolated (34.5%), followed by *Enterococcus faecium* (30%), *Escherichia coli* (26%), *Acinetobacter calcoaceticus baumannii* complex (17%), *Enterococcus faecalis* (15.5%), *Aspergillus* spp. (14%), coagulase-negative staphylococci (14%), *Enterococcus* spp. (11%), *Klebsiella pneumoniae* (9.5%), and *Staphylococcus aureus* (9.5%). When *E. cloacae* infections were examined based on whether the infections were polymicrobial (N = 84) or monomicrobial (N = 28), there was no difference in use of mechanical ventilation (68% and 61%, respectively; p = 0.583), ICU admission (83% and 71%; p = 0.170), number of operating room visits (median of 5 for both; p = 0.125), length of hospitalization (median of 56 and 58.5 days; p = 0.898), and death (5% and 4%; p = 1.00). Polymicrobial *E. cloacae* infections were further examined for 29 patients who had the combination of *E. cloacae* plus *P. aeruginosa* (with/without other pathogens). To evaluate a wider group of bacteria of high virulence, 49 patients with *E. cloacae* plus at least one bacterium of high virulence (i.e., *P. aeruginosa*, *E. coli*, *K. pneumoniae*, and/or *S. aureus*), with/without other pathogens, were assessed. All 29 patients from the *E. cloacae* plus *P. aeruginosa* combination group were also included in the 49 patients with *E. cloacae* plus bacteria of high virulence group. Use of mechanical ventilation (72% for patients with polymicrobial combination of *E. cloacae* plus *P. aeruginosa*, p = 0.183; and 73.5% for patients with polymicrobial combination of *E. cloacae* plus bacteria of high virulence, p = 0.327), ICU admission (90%, p = 0.081; and 86%, p = 0.128), length of hospitalization (median 71 days, p = 0.131; and median 67 days, p = 0.130), and death (10%, p = 0.612; and 8%, p = 0.612) were not significantly different compared to patients with monomicrobial *E. cloacae* infections. There was also no significant difference in the number of operating room visits between patients with monomicrobial *E. cloacae* infections and polymicrobial infections

**Table 3. Clinical characteristics of infected patients with single initial isolation of *E. cloacae* versus infected patients with serial isolation of *E. cloacae*.**

| Characteristic, No. (%) | Patients with single initial *E. cloacae* isolation (N = 63) | Patients with serial *E. cloacae* isolation (N = 49) | Total Patients with *E. cloacae* isolation (N = 112) | p-value |
|---|---|---|---|---|
| Age at injury, median (IQR) | 22 (21–27) | 24 (21–27) | 23 (21–27) | 0.343 |
| Male | 62 (98.4) | 49 (100) | 111 (99.1) | 1.000 |
| *Branch of Service* | | | | 0.403 |
| Air Force and Navy | 2 (3.2) | 3 (6.1) | 5 (5.5) | |
| Army | 36 (57.1) | 32 (65.3) | 68 (60.7) | |
| Marine | 25 (39.7) | 14 (28.6) | 39 (34.8) | |
| *Combat Theater* | | | | 0.441 |
| Afghanistan | 60 (95.2) | 48 (98.0) | 108 (96.4) | |
| Iraq | 3 (4.8) | 1 (2.0) | 4 (3.6) | |
| Combat Injury | 59 (93.6) | 49 (100) | 108 (96.4) | 0.073 |
| *Mechanism of Injury* | | | | 0.166 |
| Blast | 54 (85.7) | 46 (93.9) | 100 (89.3) | |
| Non-blast | 9 (14.3) | 3 (6.1) | 12 (10.7) | |
| *Blast Type* | | | | 1.00 |
| IED | 50 (79.4) | 43 (87.7) | 93 (83.0) | |
| Non-IED | 4 (6.3) | 3 (6.1) | 7 (6.2) | |
| Injured on foot patrol | 40 (63.5) | 36 (73.5) | 76 (67.9) | 0.322 |
| Burn | 10 (15.9) | 8 (16.3) | 18 (16.1) | 0.948 |
| 1st documented shock index, median (IQR) | 1.05 (0.70–1.48) | 1.16 (0.83–1.51) | 1.07 (0.74–1.49) | 0.246 |
| 1st 24 hour blood transfusion, median (IQR) | 15 (7–31) | 14 (10–31) | 15 (8–31) | 0.930 |
| *Body Region of Injury* | | | | 0.109 |
| Lower extremity | 8 (12.7) | 13 (26.5) | 21 (18.7) | |
| Upper extremity | 6 (9.5) | 1 (2.0) | 7 (6.2) | |
| Both lower and upper extremity | 48 (76.2) | 34 (69.4) | 82 (73.2) | |
| Non extremity | 1 (1.6) | 1 (2.0) | 2 (1.8) | |
| Injury severity score, median (IQR) | 34 (22–45) | 36 (27–45) | 34.5 (24–45) | 0.516 |
| *U.S. military hospital* | | | | 0.361 |
| BAMC | 19 (30.1) | 17 (34.7) | 36 (32.1) | |
| NCR | 41 (65.1) | 32 (65.3) | 73 (65.2) | |
| Both BAMC and NCR | 3 (4.8) | 0 (0) | 3 (2.7) | |
| *Mechanical ventilation* | | | | 0.872 |
| LRMC only | 9 (14.3) | 7 (14.3) | 16 (14.3) | |
| LRMC & U.S. hospital ≤1 week | 32 (50.8) | 23 (46.9) | 55 (49.1) | |
| LRMC & U.S. hospital ≥ 2 weeks | 1 (1.6) | 2 (4.1) | 3 (2.7) | |
| None | 21 (33.3) | 17 (34.7) | 38 (33.9) | |
| ICU admission | 51 (81.0) | 39 (79.6) | 90 (80.4) | 1.000 |
| Number of operating room visits, median (IQR) | 5 (4–7) | 5 (4–6) | 5 (4–6) | 0.216 |
| Polymicrobial infection[a] | 42 (66.7) | 42 (85.7) | 84 (75.0) | **0.021** |
| Hospitalization, median days (IQR) | 57 (39–88) | 57 (43–84) | 57 (40.5–84.5) | 0.904 |
| Death | 3 (4.8) | 2 (4.0) | 5 (4.5) | 1.000 |

BAMC–Brooke Army Medical Center; ICU–intensive care unit; IED–improvised explosive device; IQR–interquartile range; LRMC–Landstuhl Regional Medical Center; NCR–National Capital Region

[a] Polymicrobial infection defined as a positive culture collected within ±3 days of the *E. cloacae* culture from the same anatomical site. Organisms predominantly isolated from polymicrobial infections were *P. aeruginosa*, *E. faecium*, *E. coli*, *Acinetobacter calcoaceticus baumannii* complex, *Enterococcus faecalis*, *Aspergillus* spp. and coagulase-negative staphylococci.

**Table 4. Total duration of antibiotic use among patients with single initial isolation of *E. cloacae* versus patients with serial isolation of *E. cloacae*[a].**

| Antimicrobials | Duration of Antibiotic Use, median days (IQR) | | | p-value |
|---|---|---|---|---|
| | Patients with single initial *E. cloacae* infection (N = 63) | Patients with serial *E. cloacae* infection (N = 49) | All Patients with *E. cloacae* isolation (N = 112) | |
| Aminoglycoside | 1 (0–8) | 1 (0–4) | 1 (0–7) | 0.153 |
| Carbapenem | 12 (5–20) | 13 (4–22) | 12.5 (4–21.5) | 0.796 |
| Cephalosporin- 1st generation | 7 (3–15) | 11 (6–15) | 8.5 (4–15) | 0.052 |
| Fluoroquinolone | 7 (0–15) | 10 (3–16) | 9 (1.5–15.5) | 0.299 |
| Vancomycin | 12 (1–27) | 9 (3–28) | 10 (2–27) | 0.911 |
| Total antibiotic duration[b] | 39 (30–63) | 47 (35–72) | 43.5 (32.5–71.0) | 0.236 |

IQR–interquartile range

[a] Antibiotics that were used for a median of zero days in both groups are not shown and include aminopenicillin, anti-pseudomonal penicillin, 2nd generation cephalosporin, 3rd generation cephalosporin, 4th generation cephalosporin, clindamycin, linezolid, macrolide, monobactam, penicillin, penicillinase-resistant penicillin, polymyxin, trimethoprim-sulfamethoxazole, and topical antibiotic therapy.

[b] Total antibiotic duration was calculated as the total number of days at least one antibiotic was administered. Any days on which no antibiotics were administered are not counted in this measure.

with *E. cloacae* plus *P. aeruginosa* (median of 5 and 6, respectively, p = 0.078); however, patients with the combination of *E. cloacae* plus bacteria of high virulence had a significantly higher number of operating room visits (median 5, IQR: 5–7) compared to those with mono-microbial infections (median of 5; IQR: 4–6; p = 0.034).

## *Enterobacter cloacae* culture characteristics

All *Enterobacter* isolates were identified as *E. cloacae* (not *E. cloacae* complex). The majority of *E. cloacae* isolates were cultured from wounds (70%), followed by respiratory specimens (22%) and blood (6%) (Table 5). Seventy-five percent of the wound cultures were recovered from the lower extremities. Patients in the serial isolate group had a shorter duration from injury to *E. cloacae* isolation (median 5 days; IQR 3–13) than patients in the single initial isolate group

**Table 5. Distribution of sources of initial *E. cloacae* isolates.**

| Sites of initial *E. cloacae* culture | Initial Isolates (N = 112) |
|---|---|
| *Wound*[a] | 78 (70%) |
| Thigh | 26 (33%) |
| Lower leg | 19 (24%) |
| Pelvic, gluteal muscles, and genitalia | 8 (10%) |
| Knee | 7 (9%) |
| Foot and ankle | 7 (9%) |
| Upper arm and elbow | 5 (6%) |
| Forearm and hand | 3 (4%) |
| Head and neck | 2 (3%) |
| Abdomen | 1 (1%) |
| Respiratory | 25 (22%) |
| Blood | 7 (6%) |
| Urine | 1 (1%) |
| Intravascular Catheter Tip | 1 (1%) |

[a] The percentage for the specific wound sites is calculated using 78 as the denominator.

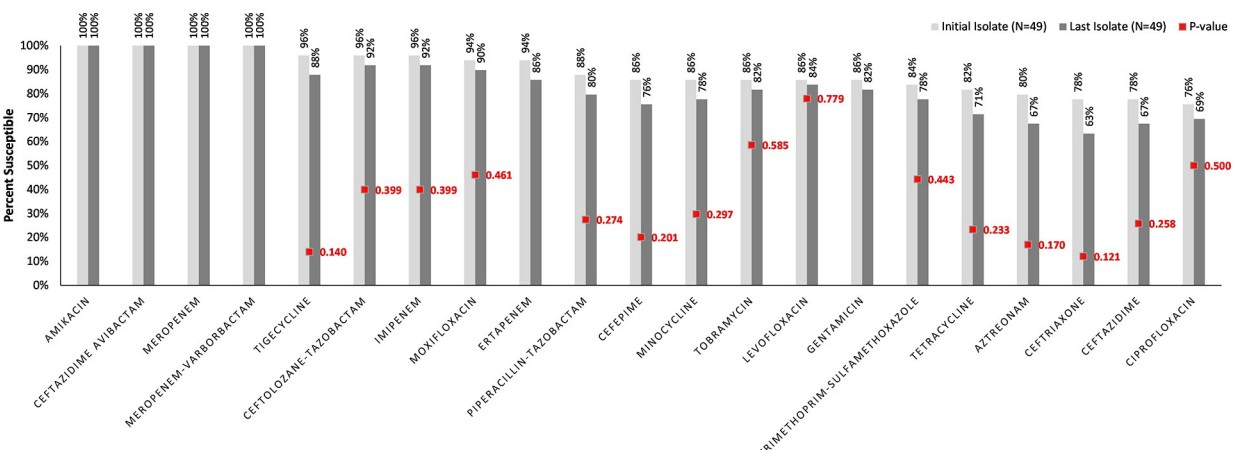

**Fig 1. Comparative antibiogram of the *E. cloacae* isolates from the 49 patients in the serial isolation group.** Ordered by decreasing susceptibilities of the initial isolate.

(median 8 days; IQR 4–7; p = 0.046). For serial *E. cloacae* patients, the median number of days between the initial isolate and last isolate was 5 days (IQR: 2–20 days).

The comparative antibiogram between the initial and last *E. cloacae* isolates in the serial isolation group is shown in Fig 1. Amikacin, ceftazidime-avibactam, meropenem, and meropenem-vaborbactam retained 100% susceptibility between initial and last isolates. The last *E. cloacae* isolates were more resistant to almost all other antibiotics. The most notable decrease in susceptibility was noted for ceftriaxone, albeit not statistically significant (78% to 63% p = 0.121). All isolates were resistant to aminopenicillins, 1st generation cephalosporins, and cephamycins.

## Discussion

To the best of our knowledge, this study is the first to specifically characterize the significance that *E. cloacae* plays in battlefield trauma-related infections and broadly compare it against other bacterial infections. Battlefield trauma patients with *E. cloacae* infections more frequently presented with complex polytrauma resulting from blast injuries than patients with non-*E. cloacae* infection. Despite their critically-ill presentations, these patients did not require greater utilization of critical care, but they experienced lengthier hospitalizations, underwent a greater number of surgical interventions, and received longer durations of antimicrobial therapy. Secondarily, while patients who had serial *E. cloacae* isolates had a shorter duration from injury to 1st infecting isolate collection than those who had single initial isolates, there were no differences in characteristics or outcomes and no significant antibiotic resistance developed in the patients from whom *E. cloacae* was recovered multiple times. Although a large proportion of the *E. cloacae* infections were polymicrobial (75% of patients), there was no difference in outcomes (e.g., ICU admission, length of hospitalization, or death) when patients with polymicrobial and monomicrobial *E. cloacae* infections were compared, including when focused on specific polymicrobial combinations of clinical relevance (i.e., *E. cloacae* plus *P. aeruginosa* and *E. cloacae* plus bacteria of high virulence). The only significant difference between the patients with monomicrobial and polymicrobial infections was an increased number of operating visits among patients with the combination of *E. cloacae* plus bacteria of high virulence (i.e., *P. aeruginosa*, *E. coli*, *K. pneumoniae*, or *S. aureus*).

*E. cloacae* is the 4[th] most common Gram-negative organism causing bloodstream infections in over 200 medical centers in 45 different nations [25]. Notably, *E. cloacae*, as well as *Klebsiella aerogenes* and *Citrobacter freundii* have been found to be the most clinically relevant AmpC producers and AmpC's conference of resistance to broad-spectrum β-lactams has been shown to produce significant adverse effects on clinical outcomes [26]. A 2002 study by Cosgrove *et al.* [27] evaluated health and economic outcomes for patients with a mean age of 63 years who had *Enterobacter* spp. infections cultured from several different anatomic sites that developed resistance to 3[rd] generation cephalosporins. Similar to other published reports [11, 12], they found that resistance developed in 10% of their population, producing an attributable longer hospital stay of 9 days, increased mortality relative risk of 5.02, and additional hospital cost of almost $30,000 [27].

In our study, compared to patients with non-*E. cloacae* infections, patients with *E. cloacae* had longer hospital stays (57 vs. 44 days) and required a significantly greater duration of antibiotic therapy (43.5 vs. 34 days); however, it should be noted that the duration of antibiotic therapy was the overall duration and not adjusted per length of hospitalization and number of operating room visits, which would impact antibiotic use (e.g., perioperative antibiotics). The comparison between patients with *E. cloacae* and non-*E. cloacae* infections did adjust for the occurrence of polymicrobial infections, including assessing clinically relevant combinations. A previous study using the TIDOS population identified that patients with *P. aeruginosa* infections had higher crude mortality compared to patients with infections attributed to other pathogens [28]. Therefore, we compared patients with monomicrobial *E. cloacae* infections to those with polymicrobial *E. cloacae* plus *P. aeruginosa* infections and there were no statistical differences in critical care or mortality between the groups. Among our total population, 17 (3% of 621) patients died and, without controlling for temporal relationship to infection or other potential factors that would potentially contribute to mortality, there was no significant mortality difference between those with and without *E. cloacae* infection (5% vs. 2%). In contrast to the 17% mortality reported by Cosgrove et al. [27], the low mortality in our patients is attributable to youth and overall better health prior to their battlefield wounds and *E. cloacae* infections.

It is noteworthy that 22% of our initial *E. cloacae* isolates were resistant to 3[rd] generation cephalosporins. Although this is lower than the prevalence of 36.4% that was seen in a national surveillance study that measured 3[rd] generation cephalosporin resistance amongst *E. cloacae* isolates cultured from American ICU patients [29], our study population's baseline *E. cloacae* resistance to 3[rd] generation cephalosporins may have contributed to the comparatively adverse outcomes seen in the *E. cloacae* infection patients compared to those with a non-*E. cloacae* infection; although, analysis of that relationship was outside the scope of this study. Despite fewer admissions to the ICU and less need for mechanical ventilation (80% vs 90% and 66% vs. 73% between the *E. cloacae* and non-*E. cloacae* infection patients, respectively), patients with *E. cloacae* infections had higher first documented shock indices (1.07 vs. 0.92), received more blood products within the first 24 hours of care (15 vs. 14 units) and had more operating room visits (5 vs. 4), which may have been secondary to the fact that patients with *E. cloacae* infection suffered more blast injuries (89% vs. 75%) and burns (16% vs. 9%) resulting in greater fluid loss [30]. Also, the greater number of burns in the *E. cloacae* infection population contributed to their significantly higher admission rate to BAMC, as BAMC is the DoD's only specialized burn center. As burns require frequent debridement, the greater number of burn injuries also potentially led to the higher number of operating room visits in the *E. cloacae* infection group.

Regarding specific antibiotic utilization, patients with *E. cloacae* infection received significantly more carbapenems, 1[st] generation cephalosporins, fluoroquinolones, and vancomycin

(Table 2). Burn injury commonly results in infection with *S. aureus*, which has led to the empiric use of vancomycin [31]. Thus, the greater number of burn injuries in the *E. cloacae* infection group, as well as the high proportion of polymicrobial infections (75% of patients) likely contributed to their greater receipt of vancomycin.

As previously mentioned, *E. cloacae* has a chromosomal AmpC β-lactamase, which is strongly induced by β-lactams, such as carbapenems and 1st generation cephalosporins. Resistance development secondary to AmpC induction is of special interest to the U.S. Armed Forces as ertapenem is recommended to be carried on the battlefield by medics in the TCCC guidelines and cefazolin is recommended as post-trauma antibiotic prophylaxis in the JTS CPG [17, 19]. These recommendations provide the possibility for the development of harmful resistance, leading to poor patient outcomes when initial wound infections are due to *E. cloacae*. As a result of the resistance-inducing pressure of carbapenems, there has been significant interest in seeking out carbapenem-sparing therapies, such as piperacillin-tazobactam or cefepime [32]. Our population did not receive a significant amount of therapy with piperacillin-tazobactam nor cefepime and the most prescribed antibiotic therapy for patients who had an *E. cloacae* infection was a carbapenem. Cefepime has also garnered significant attention as a carbapenem-sparing agent when treating AmpC-producing organisms and was only recently recommended by the IDSA as first-line therapy against *E. cloacae*, as well as *K. aerogenes* and *C. freundii* when the minimum inhibitory concentration (MIC) is known to be ≤2 µg/mL [26]. In our study, approximately 85% of isolates were susceptible to cefepime (MIC ≤2 µg/mL) with the majority having a MIC ≤0.5 µg/mL, while the resistant isolates largely had a MIC >16 µg/mL. As infection site inevitably affects clinical outcome data, it is important to note that our isolates differed from those assessed in other studies, such as the MERINO trials which focused on bacteremia [33, 34], in that our isolates were largely collected from infected traumatic wounds and only 6% of initial isolates were blood cultures. A significant proportion of the patients with *E. cloacae* infection in our study sustained burn injuries (16%), and although *Enterobacter* spp. cause a minority of burn wound infections, the bacteriology of burn wound infection remains largely Gram-negative [35].

Despite the presumed high risk of AmpC induction and de-repression given our study population's frequent receipt of carbapenems, no statistically significant resistance developed to any individual antibiotic in our study when examining serial isolates. Nevertheless, there was a non-significant trend toward resistance development against almost every β-lactam antibiotic tested except for ceftazidime-avibactam and meropenem (i.e., ceftolozane-tazobactam, imipenem, ertapenem, piperacillin-tazobactam, cefepime, aztreonam, ceftriaxone, and ceftazidime). The development of β-lactam resistance in our study is similar to prior burn wound infection literature that examined the incidence of general resistance phenotypes of *Enterobacterales* over time [36]. However, in the 2016 study by van Duin *et al*. [36], significant resistance developed over the course of weeks and in our study, the median interval between initial and last isolates was 5 days (IQR 2–20).

The lack of difference in clinical outcomes between the single initial and serial isolate groups in our study may be attributed to fast source control in both groups, as evidenced by the high number of visits to the operating room for surgical debridement. Patients in our serial *E. cloacae* group had a shorter duration to isolation after their injuries (5 vs. 8 days p = 0.046), and similar to the findings of a prior analysis of the TIDOS population [6], 75% of the *E. cloacae* infections in our analysis were associated with polymicrobial infections. Nevertheless, there was no difference in outcomes between patients in the *E. cloacae* single initial and serial isolate groups, as well as between the *E. cloacae* infection patients with polymicrobial and monomicrobial infections. Given that *E. cloacae* carries the greatest risk for AmpC derepression, our findings regarding resistance development and clinical outcomes may be

generalizable to combat trauma infections with *K. aerogenes*, *Serratia marcescens*, *C. freundii*, *Providencia stuartii*, *Morganella morganii* [37]. The lack of difference in clinical outcomes between the single initial and serial isolate groups bolsters both the DoD's current combat critical care and surgical prophylaxis guidelines with regard to AmpC induction and also supports prior literature that argued against the use of expanded Gram-negative antibiotic prophylaxis after combat trauma [17, 19, 38, 39].

Our study includes limitations inherent to retrospective studies. As our analysis was not a case-control study, the non-*E. cloacae* patients served as a comparator group rather than a control population, so matching was not applied. A potential confounder of clinical outcome differences is that a significant proportion of the patients infected with *E. cloacae* were hospitalized at BAMC, of whom, 16% were admitted for burn wound care, which likely led to prolonged hospitalizations [40]. Similar to other retrospective reports of emergence of resistance while on treatment [8, 27], we did not perform a molecular assessment to ascertain the likely mechanism for resistance observed. Even if using ceftriaxone resistance as a marker for AmpC or ESBL production, the difference in resistance between first and last isolates from the serial isolate population was not statistically significant (p = 0.121). As isolates did not undergo bacterial strain typing, we cannot directly state whether the initial and serial isolates were the same. Molecular or enzymatic characterization of β-lactamase production would likely have been useful if a significant difference in 3$^{rd}$ generation cephalosporin resistance between the initial and last isolates was detected. Lastly, the DoDTR did not capture antibiotics that were provided in the prehospital setting (e.g., ertapenem) at the time of injury, which may have limited the evaluation for β–lactam resistance development.

To the best of our knowledge, our study is the first to specifically evaluate *E. cloacae's* role as a pathogen in infection secondary to modern combat trauma. Despite DoD combat care and surgical prophylaxis guidelines recommending upfront provision of AmpC-inducing antibiotics [41], we did not see worsened clinical outcomes or significant antibiotic resistance develop in patients who experienced serial isolation of *E. cloacae* versus single initial isolation. Carbapenems were the most frequently prescribed antibiotics for our combat trauma population with *E. cloacae* infections. As IDSA and DoD guidance changes regarding antibiotic utilization, future studies on clinical outcome surveillance coupled with molecular characterization of resistance mechanisms amongst combat trauma patients are needed to ensure optimal care and support antimicrobial stewardship efforts in the Military Health System.

## Acknowledgments

We are indebted to the Infectious Disease Clinical Research Program Trauma Infectious Disease Outcomes Study team of clinical coordinators, microbiology technicians, data managers, clinical site managers, and administrative support personnel for their tireless hours to ensure the success of this project. We also wish to thank MAJ Jack Kiley for his support and guidance.

## Author Contributions

**Conceptualization:** William Bennett, Katrin Mende, Wesley R. Campbell, David R. Tribble, Joseph M. Yabes.

**Data curation:** William Bennett, Katrin Mende, Miriam Beckius, Joseph M. Yabes.

**Formal analysis:** William Bennett, Katrin Mende, Miriam Beckius, Laveta Stewart, Faraz Shaikh, Azizur Rahman, Joseph M. Yabes.

**Investigation:** William Bennett, Katrin Mende, Wesley R. Campbell, Laveta Stewart, Faraz Shaikh, Azizur Rahman, David R. Tribble, Joseph M. Yabes.

**Writing – original draft:** William Bennett.

**Writing – review & editing:** Katrin Mende, Wesley R. Campbell, Miriam Beckius, Laveta Stewart, Faraz Shaikh, Azizur Rahman, David R. Tribble, Joseph M. Yabes.

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
