## [Decision Letter · Decision Letter 0]

24 May 2023

PONE-D-23-05801Enterobacter cloacae infection characteristics and outcomes in battlefield trauma patientsPLOS ONE

Dear Dr. Bennett,

Thank you for submitting your manuscript to PLOS ONE. After careful consideration, we feel that it has merit but does not fully meet PLOS ONE’s publication criteria as it currently stands. Therefore, we invite you to submit a revised version of the manuscript that addresses the points raised during the review process.

We look forward to receiving your revised manuscript.

Kind regards,

Dona Benadof, M.D

Academic Editor

PLOS ONE

Journal Requirements:

Additional Editor Comments:

It is an interesting text, especially in the context that it refers to an area in which there is little literature, such as war wounds. I suggest reviewing the comments of the reviewers, and especially those of the microbiological field, since they are relevant to clarify possible confusions in the clinical field regarding patterns of susceptibility of microorganisms, relevance of the antimicrobials mentioned in the paper and resistance mechanisms of the microorganisms mentioned. I suggest review by a clinical microbiologist.And include the limitations sugested by reviewers.

Reviewers' comments:

Reviewer's Responses to Questions

**Comments to the Author**

1. Is the manuscript technically sound, and do the data support the conclusions?

Reviewer #1: Partly

Reviewer #2: Partly

Reviewer #3: Yes

2. Has the statistical analysis been performed appropriately and rigorously? 

Reviewer #1: No

Reviewer #2: Yes

Reviewer #3: Yes

3. Have the authors made all data underlying the findings in their manuscript fully available?

Reviewer #1: Yes

Reviewer #2: Yes

Reviewer #3: No

4. Is the manuscript presented in an intelligible fashion and written in standard English?

Reviewer #1: Yes

Reviewer #2: Yes

Reviewer #3: Yes

5. Review Comments to the Author

Reviewer #1: Bennett and colleagues present a manuscript describing patients with E. cloacae infections after suffering battlefield trauma.

The subject of the manuscript is interesting and the authors are right when they stress the increasing relevance of infections due to MDR E. cloacae in different clinical settings, including trauma patients. However, there are a number of issues that require attention before publication.

Major comments:

1. In the introduction, the authors make the point of a possible causal effect between Enterobacter infections and the development of other "MDRGN infections". In the same line, it is suggested that "earlier identification and treatment of Enterobacter spp. infections may reduce the acquisition of secondary MDRGN infections". However, no data are provided this claim. I think throughout the manuscript there is a confusion between the possibility of developing resistance due to AmpC induction (that is in a same Enterobacter isolate previously exposed to the appropriate environmental conditions) and the risk of developing infections due to other MDR organisms. Also, the concept of 'induction' on resistance is never really addressed.

2. The Aim of the authors is to study “the epidemiological characteristics of patients with E. cloacae infections, characterized the antibiotic prescribing patterns used in the treatment of these infections and their effects on developing resistance, and examined clinical outcomes compared to battlefield trauma patients who developed bacterial infections associated with organisms other than E. cloacae. Also they “described clinical characteristics and outcome differences between patients with solitary isolation of E. cloacae and no known microbiological relapse and patients with serial isolation to assess for antimicrobial resistance development.” While all these questions are interesting, the study design has several problems that require attention. It is not clear to me how was infection defined, which is a major issue considering most of the cultures come from non-sterile sites such as wounds. Were all these cultures obtained from tissue under sterile conditions? Were swabs accepted? One example of the potential importance of this issue is that while coagulase negative staphylococci were the most frequent organisms in the non-Enterobacter group, the median days of vancomycin use for that group was 0. Moreover, the statement provided in lines 137-141 further emphasizes my point, where the authors conveniently chose isolates “more likely to be a true infection”. This is a fundamental issue that requires to be thoroughly addressed.

3. The definition of solitary is also confusing and sometimes blurred with monomicrobial vs. polymicrobial. Please clarify. How did you define relapse vs. persistent infection? Also, the fact that there is no molecular epidemiology studies precludes making any strong conclusion regarding subsequent isolates as there is no strong data proving isolates are actually the same (or even related)

4. After dealing with the issues of inclusion criteria stated above, the statistical analyses need to be improved to address possible confounding factors and assess the strength of the associations.

Minor comments:

1. Lines 70-75. Please italicize Enterobacteriaceae (ideally change to the updated taxonomical denomination of Enterobacterales). Also, please provide a reference for foe the statement ending in line 72.

2. Line 76. When you say “have been found to develop β-lactam resistance during treatment”, are you referring to 3rd gen cephalosporins? Please clarify and provide a reference.

3. Please rephrase lines 154-155.

Reviewer #2: Thank you for your submission. Please find below some comments about your manuscript.

1. It is important to harmonize if the isolates belong to Enterobacter cloacae complex. If so, please specify in all the manuscript E. cloacae complex.

2. Page 5 line 80. Please specify if it is amoxicillin/clavulanic acid as it seems separated by coma.

3. Please include the method used to identify the isolates. Was it MALDI TOF ? or just Phoenix 100 ?.

4. If there were not typing available to compare the first and subsequent isolates ( solitary vs serial ), it should be included as a limitation. There is a possibility of different isolates of E. cloacae in the same patient.

5. I suggest avoiding the report 0% of susceptibility for drugs with intrinsic resistance like cefoxitin, 1st and 2nd gen cephalosporins, amoxicillin/clavulanic, ampicillin, etc.

6. I suggest including also the IDSA guidance recommendations in the discussion. About the importance of E. cloacae along with C. freundii and K. aerogenes as derepression of AmpC may occur.

7. Please include or mention the 3rd generation cephalosporin used. Is it ceftriaxone?.

8. Page 22 line 385. Please change Enterobacteriaceae by Enterobacterales.

9. Please considering the reports, propose an antimicrobial stewardship strategy. For example include rapid ID diagnosis to switch from 1st gen cephalosporins to cefepime or ertapenem.

10. Please include the CLSI breakpoints used to interpret AST results. For example M100 33rd 2023.

11. In figure 1, I suggest to delete nitrofurantoin as the manuscript refers to soft tissue and bone infections. Also, cefuroxime should be deleted as is intrinsically resistant. Ceftaroline is not the preferred drug for E. cloacae. It may be removed also.

12. If possible, I suggest including MIC distributions for cefepime.

Reviewer #3: This is an interesting manuscript reporting infections due to Enterobacter claocae in patient with wounds acquired in battlefield. This is a retrospective, descriptive study. The authors compare mainly outcomes in this group of patients with E. cloacae infections and patient’s infections without E cloacae. Most of the infections are in wounds due to blast or burn. Most of the infections in both groups were polymicrobial.

This manuscript is well written. It follows the editorial guidelines. The title reflects the content. Abstract is correct. Introduction is too extensive. I recommend to shorter it. Methodology is of simple clinical series and with a comparator.

Results: since 75% of the infections are polymicrobial, I would like to have a better description of the combination of bacteria in both groups, in order to have a better understanding of the severity of infections and antibiotic treatments and outcomes. While 66.7 % of the patients in the E. cloacae group are polymicrobial, the role of E. cloacae is questionable in different outcomes, especially in wounds. In addition, severity of illness, complications and many interventions could be more related to the trauma injury rather the role of the infection. It is of interest to see how the susceptibility profile does not change much over time in these isolates of E. cloacae.

Discussion is also too extensive. A shorter discussion would be better. In discussion the authors state that there were no differences when compared monomicrobial with polymicrobial infections. This information is not in the results. Also, it is not clear if the comparison is in the E. cloacae infections group or in the when compared with non E. cloacae infections.

Table 1 and figure 1 are fine. Table 2 and 3 are not necessary since the outcomes are not relevant. Duration of antibiotics, in spite there are differences, is not relevant, since the duration of treatment is not prolonged in most patients. References are fine.

In summary, the manuscript is well written, methods are fine. The results are reliable since are based in microbiological and clinical feature. In addition, the conclusion is based in the results. However, most of the infections are polymicrobial, and the role of E. cloacae in these infections is not clear and are of not enough interest. A better description of the polymicrobial infections and the combination of treatment is needed to have a better understanding and see the importance of this study. In addition, these missing data could help to understand why they study E. cloacae, in step of S. aureus or Pseudomonas aeruginosa, or other specific bacteria in these infections. This data could help to support the study and see the role of E. cloacae in these infections. Finally, this manuscript could fit better as a brief report.

6. PLOS authors have the option to publish the peer review history of their article (what does this mean?). If published, this will include your full peer review and any attached files.

Reviewer #1: No

Reviewer #2: No

Reviewer #3: No

---

## [Author Response · Author response to Decision Letter 0]

5 Jul 2023

RESPONSE TO REVIEWERS

Journal Requirements:

Author Response: The manuscript has been formatted per PLOS ONE’s style requirements, including file naming.

Author Response. The following information is included as the last sentence of the 1st paragraph of the Methods section and in the online submission form: ‘Data and specimens were collected from individuals who provided authorization through informed consent and HIPAA authorization processes, or through an IRB-approved waiver of consent for use of de-identified data not obtained through interaction or intervention with human subjects.’

Author Response: All relevant data are included in the manuscript and tables/figures. Per our organization’s Institutional Review Board, we do have restrictions on making the data publicly available. As such, we have modified the Data Availability statement to the following: 

‘Data for this study are available from the Infectious Disease Clinical Research Program (IDCRP), headquartered at the USU, Department of Preventive Medicine and Biostatistics. Review by the USU Institutional Review Board and approval of data sharing agreements are required for use of the data collected under this protocol. Data requests may be sent to: Address: 6270A Rockledge Drive, Suite 250, Bethesda, MD 20817. Email: contactus@idcrp.org.’ 

Author Response: There are restrictions placed on making the data publicly available per our Institutional Review Board and requests for use must be reviewed by them.

Additional Editor Comments:

It is an interesting text, especially in the context that it refers to an area in which there is little literature, such as war wounds. I suggest reviewing the comments of the reviewers, and especially those of the microbiological field, since they are relevant to clarify possible confusions in the clinical field regarding patterns of susceptibility of microorganisms, relevance of the antimicrobials mentioned in the paper and resistance mechanisms of the microorganisms mentioned. I suggest review by a clinical microbiologist. And include the limitations suggested by reviewers.

Author Response: Thank you for your comments. We have reviewed the comments from the peer reviewers and revised the manuscript accordingly, including adding an additional statement to the limitations section. One of the co-authors is a microbiologist and she had comprehensively reviewed the reviewer comments and worked with the lead authors on the revision of the manuscript. 

REVIEWER COMMENTS’

Reviewer's Responses to Questions

Comments to the Author

1. Is the manuscript technically sound, and do the data support the conclusions?

Reviewer #1: Partly

Reviewer #2: Partly

Reviewer #3: Yes

2. Has the statistical analysis been performed appropriately and rigorously?

Reviewer #1: No

Reviewer #2: Yes

Reviewer #3: Yes

3. Have the authors made all data underlying the findings in their manuscript fully available?

Reviewer #1: Yes

Reviewer #2: Yes

Reviewer #3: No

4. Is the manuscript presented in an intelligible fashion and written in standard English?

Reviewer #1: Yes

Reviewer #2: Yes

Reviewer #3: Yes

5. Review Comments to the Author

REVIEWER #1: 

Bennett and colleagues present a manuscript describing patients with E. cloacae infections after suffering battlefield trauma. The subject of the manuscript is interesting and the authors are right when they stress the increasing relevance of infections due to MDR E. cloacae in different clinical settings, including trauma patients. However, there are a number of issues that require attention before publication.

Major comments:

1. In the introduction, the authors make the point of a possible causal effect between Enterobacter infections and the development of other "MDRGN infections". In the same line, it is suggested that "earlier identification and treatment of Enterobacter spp. infections may reduce the acquisition of secondary MDRGN infections". However, no data are provided this claim. I think throughout the manuscript there is a confusion between the possibility of developing resistance due to AmpC induction (that is in a same Enterobacter isolate previously exposed to the appropriate environmental conditions) and the risk of developing infections due to other MDR organisms. Also, the concept of 'induction' on resistance is never really addressed.

Author Response: Thank you for your comments. We have revised the text in the Introduction to remove the language suggesting that Enterobacter spp. infections may have contributed to acquisition of secondary MDRGN infections. The revised text (lines 59-64) reads ‘In that analysis, Enterobacter spp. was the second most commonly isolated Gram-negative organism from polymicrobial cultures and was noted to have a shorter time from injury to first infection. As polymicrobial infections among blast wound infections were more likely to produce MDR organisms and Enterobacter spp. infections have a potentially quicker onset, earlier identification and treatment of Enterobacter spp. infections may improve patient outcomes.’

Regarding your comments about AmpC induction, we agree that there may have been more mention of AmpC induction in the Introduction than needed, considering that we were unable to directly report on it in the Results. As it is an important mechanism related to resistance of E. cloacae, we did not remove the topic from the Introduction, but we revised the text to reduce its mention (deleted two sentences and revised text on lines 78-80). The sentence on lines 84-86 was also revised to indicate that the focus is more on increasing beta-lactam resistance versus AmpC induction. A sentence was also added to the end of the Laboratory Analysis section of the Methods (lines 142-144) to state that ‘As molecular assays to assess for AmpC were not routinely conducted at the military hospitals during the study period, data on AmpC induction were not available.’ 

While we do not have data from the molecular assays, we have addressed the concept of AmpC induction on resistance in the Discussion to our best ability via assessment of the initial and final isolates’ antibiograms (lines 431-433). The revised text reads ‘Even if using ceftriaxone resistance as a marker for AmpC or ESBL production, the difference in resistance between first and last isolates from the serial isolate population was not statistically significant (p=0.121). As isolates did not undergo bacterial strain typing, we cannot directly state whether the initial and serial isolates were the same. Molecular or enzymatic characterization of β-lactamase production would likely have been useful if a significant difference in 3rd generation cephalosporin resistance between the initial and last isolates was detected.’

2. The Aim of the authors is to study “the epidemiological characteristics of patients with E. cloacae infections, characterized the antibiotic prescribing patterns used in the treatment of these infections and their effects on developing resistance, and examined clinical outcomes compared to battlefield trauma patients who developed bacterial infections associated with organisms other than E. cloacae. Also they “described clinical characteristics and outcome differences between patients with solitary isolation of E. cloacae and no known microbiological relapse and patients with serial isolation to assess for antimicrobial resistance development.” While all these questions are interesting, the study design has several problems that require attention. It is not clear to me how was infection defined, which is a major issue considering most of the cultures come from non-sterile sites such as wounds. Were all these cultures obtained from tissue under sterile conditions? Were swabs accepted? One example of the potential importance of this issue is that while coagulase negative staphylococci were the most frequent organisms in the non-Enterobacter group, the median days of vancomycin use for that group was 0. Moreover, the statement provided in lines 137-141 further emphasizes my point, where the authors conveniently chose isolates “more likely to be a true infection”. This is a fundamental issue that requires to be thoroughly addressed.

Author Response: Thank you for your questions. Infections were defined using standardized criteria from the CDC National Healthcare Safety Network, as described on lines 121-124. Isolates were classified as infecting if they were collected from clinical work-ups per suspicion of infection.

Regarding your question if swabs were collected, the majority were operative cultures. As E. cloacae was the focus of this paper, sterile body sites were considered to be a reliable source for cultures for infecting isolates when multiple isolates were available from the first day of isolation from different sources. This is substantiated by the findings of Pien et al. (Am J Med. 2010;123(9):819-828), which evaluated 2,669 isolates from 2,273 positive blood cultures and determined that 51% of those isolates represented true infections. Among the E. cloacae isolates identified in the Pien et al. study, 93% were considered true bloodstream infections. In our study, there was a low occurrence of patients with multiple E. cloacae isolates collected on the first day of isolation and from those patients, 8 of the selected E. cloacae isolates were cultured from sterile body sites. 

3. The definition of solitary is also confusing and sometimes blurred with monomicrobial vs. polymicrobial. Please clarify. How did you define relapse vs. persistent infection? Also, the fact that there is no molecular epidemiology studies precludes making any strong conclusion regarding subsequent isolates as there is no strong data proving isolates are actually the same (or even related)

Author Response: Thank you for your comments and we agree that the term ‘solitary’ may be confusing. We have revised the text throughout to replace ‘solitary’ with ‘single initial’ and clarified that the definition refers only to the isolation of E. cloacae and does not preclude the cultures being polymicrobial. The revised text on lines 124-127 reads ‘Inclusion criteria for the single initial E. cloacae isolate group required patients to have an infecting E. cloacae isolate collected from the initial culture (may be either monomicrobial or polymicrobial) with no isolation from subsequent cultures.’ 

Regarding relapse vs persistent infection, we did not differentiate relapse vs persistent infection; however, our underlying hypothesis of differing clinical characteristics between patients with serial vs single initial E. cloacae isolation was based on the fact that patients with single initial E. cloacae isolation did not have known “relapse” or “persistent infection” on the basis of repeated culture.

We agree that the lack of molecular analysis is a limitation of the study and have added a sentence to the limitations paragraph in the Discussion (lines 433-435) to read ‘As isolates did not undergo bacterial strain typing, we cannot directly state whether the initial and serial isolates were the same.’

4. After dealing with the issues of inclusion criteria stated above, the statistical analyses need to be improved to address possible confounding factors and assess the strength of the associations.

Author Response: Thank you for your suggestion. However, we feel that our inclusion criteria are appropriate to the analysis (please see the responses above). The potential of confounding from polymicrobial infections among the patients with E. cloacae infections is addressed in the Results text with expanded analysis included per the comments from Reviewer #3. As the purpose of the analysis was to delineate the epidemiology of E. cloacae infections in battlefield trauma, we feel that the univariate analyses included in the Results are appropriate for the study. Based on the findings in the study, multivariate modeling to identify risk factors for infections or predictors of poor outcomes may be warranted, but that is outside the scope of this study.

Minor comments:

1. Lines 70-75. Please italicize Enterobacteriaceae (ideally change to the updated taxonomical denomination of Enterobacterales). Also, please provide a reference for foe the statement ending in line 72.

Author Response: Thank you for the suggestion. Enterobacteriaceae has been changed to Enterobacterales and italicized in the text. A reference (new #7) was added to the statement per you indicated.

2. Line 76. When you say “have been found to develop β-lactam resistance during treatment”, are you referring to 3rd gen cephalosporins? Please clarify and provide a reference.

Author Response: Thank you for your question. The sentence has been revised to be ‘E. cloacae is initially phenotypically susceptible to 3rd generation cephalosporins (e.g., ceftriaxone and ceftazidime) in vitro; however, as high as 19% of isolates developed β-lactam resistance during treatment [9-12].’

3. Please rephrase lines 154-155.

Author Response: Thank you for the suggestion. The sentence was revised to the following: ‘Categorical variables were assessed using �2 and Fisher’s Exact Tests, where appropriate.’

REVIWER #2

Thank you for your submission. Please find below some comments about your manuscript.

1. It is important to harmonize if the isolates belong to Enterobacter cloacae complex. If so, please specify in all the manuscript E. cloacae complex.

Author Response: Thank you for the question. The BD Phoenix Automated Microbiology System used in this analysis does have the option to identify isolates as E. cloacae complex; however, all the isolates included in our study were specifically identified as E. cloacae. A new sentence (line 281) was added to the Enterobacter Cloacae Culture Characteristics section of the Results to state ‘All Enterobacter isolates were identified as E. cloacae (not E. cloacae complex).’

2. Page 5 line 80. Please specify if it is amoxicillin/clavulanic acid as it seems separated by comma.

Author Response: Thank you for catching that typo. The text was revised to amoxicillin/clavulanic acid.

3. Please include the method used to identify the isolates. Was it MALDI TOF ? or just Phoenix 100 ?.

Author Response: Thank you for the question. The method used to identify the isolates was the BD Phoenix Automated Microbiology System. The first sentence in the Methods Laboratory Analysis section was revised to clarify this point. It now reads ‘Identification and susceptibility testing of the E. cloacae isolates were performed using the BD Phoenix Automated Microbiology System (NMIC/ID-308 and NMIC-311 panels, BD Diagnostics, Sparks, MD).’ 

4. If there were not typing available to compare the first and subsequent isolates ( solitary vs serial ), it should be included as a limitation. There is a possibility of different isolates of E. cloacae in the same patient.

Author Response: Thank you for the suggestion and we agree that this is a limitation of our analysis and have added a statement to the limitations paragraph in the Discussion (lines 433-435) to read ‘As isolates did not undergo bacterial strain typing, we cannot directly state whether the initial and serial isolates were the same.’.

5. I suggest avoiding the report 0% of susceptibility for drugs with intrinsic resistance like cefoxitin, 1st and 2nd gen cephalosporins, amoxicillin/clavulanic, ampicillin, etc.

Author Response: Thank you for the suggestion. The statement in the legend of Figure 1 reporting the 0% susceptibility for the antibiotics with intrinsic resistance has been deleted.

6. I suggest including also the IDSA guidance recommendations in the discussion. About the importance of E. cloacae along with C. freundii and K. aerogenes as derepression of AmpC may occur.

Author Response: Thank you for the suggestion. The sentence on lines 323-326 in the Discussion has been revised to state ‘Notably, E. cloacae, as well as Klebsiella aerogenes and Citrobacter freundii have been found to be the most clinically relevant AmpC producers and AmpC’s conference of resistance to broad-spectrum β-lactams has been shown to produce significant adverse effects on clinical outcomes.’ In addition, we have also revised the sentence on lines 384-387 to read ‘Cefepime has also garnered significant attention as a carbapenem-sparing agent when treating AmpC-producing organisms and was only recently recommended by the IDSA as first-line therapy against E. cloacae, as well as K. aerogenes and C. freundii when the minimum inhibitory concentration (MIC) is known to be ≤2 µg/mL.’

7. Please include or mention the 3rd generation cephalosporin used. Is it ceftriaxone?.

Author Response: Thank you for the question. However, we do not have further specificity with regard to the 3rd generation cephalosporin as that is how it was recorded and both ceftriaxone and ceftazidime were options available through the DoD. Resistance amongst initial E. cloacae isolates was 22% to both ceftriaxone and ceftazidime with 63% and 67% resistance in the last isolate group, respectively.

8. Page 22 line 385. Please change Enterobacteriaceae by Enterobacterales.

Author Response: Thank you for the suggestion. We have made that change in the text.

9. Please considering the reports, propose an antimicrobial stewardship strategy. For example include rapid ID diagnosis to switch from 1st gen cephalosporins to cefepime or ertapenem.

Author Response: Thank you for the suggestion. The purpose of this study was not to develop recommendations or propose practice changes for the DoD, but to characterize the epidemiology of these infections. We also feel that our findings at this time do not justify making direct recommendations as there were no statistically significant differences in outcomes identified between the patients who had single initial isolation versus serial isolation of E. cloacae. Nevertheless, antimicrobial stewardship is an important priority for the Military Health System and our findings will be shared with individuals focused on improving antimicrobial stewardship at clinical sites. The last sentence (lines 446-449) in the conclusions was revised to incorporate the need to support antimicrobial stewardship efforts in the Military Health System and reads ‘As IDSA and DoD guidance changes regarding antibiotic utilization, future studies on clinical outcome surveillance coupled with molecular characterization of resistance mechanisms amongst combat trauma patients are needed to ensure optimal care and support antimicrobial stewardship efforts in the Military Health System.’ 

10. Please include the CLSI breakpoints used to interpret AST results. For example M100 33rd 2023.

Author Response: Thank you for the question. The AST results were interpreted using the most updated BD Phoenix software at that time which was in concordance with the CLSI M100-S30 Performance Standards for Antimicrobial Susceptibility Testing - 30th Info Supplement – 2020. The sentence in the Results Laboratory Analysis section was revised to clarify that point and the CLSI volume is included as citation #24. The sentence now reads ‘Antimicrobial susceptibility testing results were interpreted in accordance with the Clinical Laboratory Standards Institute (CLSI M100 30th edition) breakpoints to construct an antibiogram [24].’

11. In figure 1, I suggest to delete nitrofurantoin as the manuscript refers to soft tissue and bone infections. Also, cefuroxime should be deleted as is intrinsically resistant. Ceftaroline is not the preferred drug for E. cloacae. It may be removed also.

Author Response: Thank you for the suggestion. As this is a descriptive study and colonization by these organisms may predispose to other mechanisms of infection, we would prefer to keep all the antimicrobials in Figure 1 so it is a broad-ranging antibiogram. 

12. If possible, I suggest including MIC distributions for cefepime.

Author Response: Thank you for the suggestion. A statement was added to the Discussion on lines 387-389 that reads ‘In our study, approximately 85% of isolates were susceptible to cefepime (MIC ≤2 µg/mL) with the majority having a MIC ≤0.5 µg/mL, while the resistant isolates largely had a MIC >16 µg/mL.’

REVIWER #3: 

This is an interesting manuscript reporting infections due to Enterobacter claocae in patient with wounds acquired in battlefield. This is a retrospective, descriptive study. The authors compare mainly outcomes in this group of patients with E. cloacae infections and patient’s infections without E cloacae. Most of the infections are in wounds due to blast or burn. Most of the infections in both groups were polymicrobial.

This manuscript is well written. It follows the editorial guidelines. The title reflects the content. Abstract is correct. Introduction is too extensive. I recommend to shorter it. Methodology is of simple clinical series and with a comparator.

Author Response: Thank you for your comments. We feel that the information included in the Introduction was important for a reader to understand the relevance regarding why our study was needed; however, we do see your point and have revised the text to slightly shorten it. 

Results: since 75% of the infections are polymicrobial, I would like to have a better description of the combination of bacteria in both groups, in order to have a better understanding of the severity of infections and antibiotic treatments and outcomes. While 66.7 % of the patients in the E. cloacae group are polymicrobial, the role of E. cloacae is questionable in different outcomes, especially in wounds. In addition, severity of illness, complications and many interventions could be more related to the trauma injury rather the role of the infection. It is of interest to see how the susceptibility profile does not change much over time in these isolates of E. cloacae.

Author Response: Thank you for your comments. We agree that it is challenging to determine the impact of specific bacteria isolated from polymicrobial cultures on outcomes. After identifying differences in outcomes between patients with E. cloacae infections versus those with non-E. cloacae infection, we did further analysis of the E. cloacae group to try and better delineate the role of E. cloacae with regard to outcomes. Specifically, we compared requirements for mechanical ventilation, ICU admission, number of operating room visits, length of hospitalization, and death between patients with monomicrobial E. cloacae infections and polymicrobial E. cloacae infections and identified no significant differences between the groups (please see lines 229-234 in the Results text). 

We thank you for your suggestion to further examine the polymicrobial combinations. In a previous study using the TIDOS population, we examined characteristics of patients with Pseudomonas aeruginosa infections and identified that those patients had significantly higher crude mortality compared to patients with infections attributed to other pathogens (Ford et al. Mil Med. 2022; 187(3/4):426-434). Based on that finding and similar observations in the literature, as well as the high frequency of patients with P. aeruginosa isolated in the polymicrobial infections, we assessed characteristics of 29 patients with the polymicrobial combination of E. cloacae plus P. aeruginosa, as well as 49 patients with the combination of E. cloacae plus bacteria of high virulence (i.e., P. aeruginosa, E. coli, K. pneumoniae, and/or S. aureus), and compared characteristics against those of the patients with monomicrobial E. cloacae infections. Similar to the findings from the comparison with the overall polymicrobial group, there were no significant differences in the characteristics between the patients with monomicrobial E. cloacae infections and the polymicrobial combination of E. cloacae plus P. aeruginosa. When patients with the polymicrobial combination of E. cloacae plus bacteria of high virulence were compared to monomicrobial infections, the only significant difference was that the polymicrobial patients had a greater number of operating room visits. These findings have been added to the Results text on lines 234-251 and also incorporated into the Discussion on lines 316-321 and lines 337-344. 

While the frequently identified organisms from polymicrobial infections are listed as footnotes with Tables 1 and 3, we have added a new sentence (lines 225-229) to provide the distribution of other organisms isolated from patients with the polymicrobial infections. The sentence reads ‘Among the 84 patients with polymicrobial infections, Pseudomonas aeruginosa was the most frequently isolated (34.5%), followed by Enterococcus faecium (30%), Escherichia coli (26%), Acinetobacter calcoaceticus baumannii complex (17%), Enterococcus faecalis (15.5%), Aspergillus spp. (14%), coagulase-negative staphylococci (14%), Enterococcus spp. (11%), Klebsiella pneumoniae (9.5%), and Staphylococcus aureus (9.5%).’ 

Discussion is also too extensive. A shorter discussion would be better. In discussion the authors state that there were no differences when compared monomicrobial with polymicrobial infections. This information is not in the results. Also, it is not clear if the comparison is in the E. cloacae infections group or in the when compared with non E. cloacae infections.

Author Response: Thank you for your comments. We have made revisions to shorten the Discussion slightly.

The comparison of monomicrobial and polymicrobial infections is specific to the individuals with E. cloacae infections. The text in the Results section comparing the monomicrobial and polymicrobial E. cloacae infections can be found on lines 229-234. The text reads ‘When E. cloacae infections were examined based on whether the infections were polymicrobial (N=84) or monomicrobial (N=28), there was no difference in use of mechanical ventilation (68% and 61, respectively; p=0.583), ICU admission (83% and 71%; p=0.170), number of operating room visits (median of 5 for both; p=0.125), length of hospitalization (median of 56 and 58.5 days; p=0.898), and death (5% and 4%; p=1.00).’ Per your suggestion, we have also added data from further examination of the E. cloacae polymicrobial combinations and a statement with data on the distribution of other organisms in the polymicrobial E. cloacae infections (see response above and lines 234-251 and 225-229 in the revised manuscript).

Regarding the sentence in the Discussion (lines 413-416), to clarify that the monomicrobial and polymicrobial comparisons were within the E. cloacae infection group, the sentence was revised to the following: ‘Nevertheless, there was no difference in outcomes between patients in the E. cloacae single initial and serial isolate groups, as well as between the E. cloacae infection patients with polymicrobial and monomicrobial infections.’ 

Table 1 and figure 1 are fine. Table 2 and 3 are not necessary since the outcomes are not relevant. Duration of antibiotics, in spite there are differences, is not relevant, since the duration of treatment is not prolonged in most patients. References are fine.

Author Response: Thank you for your comments. We feel that the data included in Tables 2 and 3 are valuable for the reader. While the duration of treatment was not prolonged, we believe that readers would like to know details on the duration of treatment as it potentially relates to changes in susceptibility with the serial isolates. We also feel that it is important to have Table 3 to show that there were no statistical differences between patients who had only single initial versus serial isolation of E. cloacae as prolonged isolation of other pathogens have been attributed to worsened clinical status and outcomes. 

In summary, the manuscript is well written, methods are fine. The results are reliable since are based in microbiological and clinical feature. In addition, the conclusion is based in the results. However, most of the infections are polymicrobial, and the role of E. cloacae in these infections is not clear and are of not enough interest. A better description of the polymicrobial infections and the combination of treatment is needed to have a better understanding and see the importance of this study. In addition, these missing data could help to understand why they study E. cloacae, in step of S. aureus or Pseudomonas aeruginosa, or other specific bacteria in these infections. This data could help to support the study and see the role of E. cloacae in these infections. Finally, this manuscript could fit better as a brief report.

Author Response: Thank you for your comments. We agree that the description of the polymicrobial infections is very important to our conclusions and we do have data in the text comparing characteristics between the patients with E. cloacae monomicrobial and polymicrobial infections (lines 229-234). Per your suggestion, we have expanded the text to compare patients with monomicrobial E. cloacae infections to those with specific polymicrobial combinations (i.e., E. cloacae plus P. aeruginosa and E. cloacae plus bacteria of high virulence; please see response above and lines 234-251 and 225-229 in the text).

---

## [Editor Report · Decision Letter 1]

8 Aug 2023

PONE-D-23-05801R1Enterobacter cloacae infection characteristics and outcomes in battlefield trauma patientsPLOS ONE

Dear Dr. William Bennett

Thank you for submitting your manuscript to PLOS ONE. After careful consideration, we feel that it has merit but does not fully meet PLOS ONE’s publication criteria as it currently stands. Therefore, we invite you to submit a revised version of the manuscript that addresses the points raised during the review process.

ACADEMIC EDITOR:*You have done a good job with the suggestions and changes of manuscript. *

 I suggest a minor change that implies taking the reviewer's suggestion that I indicate below: "In figure 1, I suggest to delete nitrofurantoin as the manuscript refers to soft tissue and bone infections. Also, cefuroxime should be deleted as is intrinsically resistant. Ceftaroline is not the preferred drug for E. cloacae. It may be removed also."  Despite the fact that you indicate and justify it as a broad-spectrum antibiogram, in reality clinical microbiology should inform the antibiogram according to the identified microorganism and the site of infection. I think that this figure can be an element of confusion for the reader.==============================

We look forward to receiving your revised manuscript.

Kind regards,

Dona Benadof, M.D

Academic Editor

PLOS ONE
---

## [Author Response · Author response to Decision Letter 1]

11 Aug 2023

ACADEMIC EDITOR:

You have done a good job with the suggestions and changes of manuscript.

Author Response: Thank you very much.

I suggest a minor change that implies taking the reviewer's suggestion that I indicate below:

"In figure 1, I suggest to delete nitrofurantoin as the manuscript refers to soft tissue and bone infections. Also, cefuroxime should be deleted as is intrinsically resistant. Ceftaroline is not the preferred drug for E. cloacae. It may be removed also."

Despite the fact that you indicate and justify it as a broad-spectrum antibiogram, in reality clinical microbiology should inform the antibiogram according to the identified microorganism and the site of infection. I think that this figure can be an element of confusion for the reader.

Author Response: Thank you for your suggestion. Figure 1 has been modified per your comments and ceftaroline has been removed from the sentence on lines 399-403.

Journal Requirements:

Author Response: Thank you. The reference list has been reviewed and it is complete. There was a change for the URL for reference #19, which was updated in the revised version. Reference #26 was also updated to reflect the current version available for that document. The rest of the references are correct. None of the papers cited have been retracted.

---

## [Editor Report · Decision Letter 2]

15 Aug 2023

Enterobacter cloacae infection characteristics and outcomes in battlefield trauma patients

PONE-D-23-05801R2

Dear Dr.William Bennet

We’re pleased to inform you that your manuscript has been judged scientifically suitable for publication and will be formally accepted for publication once it meets all outstanding technical requirements.

Kind regards,

Dona Benadof, M.D

Academic Editor

PLOS ONE

---

## [Editor Report · Acceptance letter]

21 Aug 2023

PONE-D-23-05801R2 

*Enterobacter cloacae* infection characteristics and outcomes in battlefield trauma patients 

Dear Dr. Bennett:

I'm pleased to inform you that your manuscript has been deemed suitable for publication in PLOS ONE. Congratulations! Your manuscript is now with our production department. 

Kind regards, 

on behalf of

Dr. Dona Benadof 

Academic Editor

PLOS ONE